# Circuit Design of a Seven-piecewise Linear Activation Function

1st Ren Cai
*School of Electrical and Information Engineering,*
*Wuhan Institute of Technology*
Wuhan, China
cairen_stu@163.com

2nd Le Yang*
*School of Electrical and Information Engineering,*
*Wuhan Institute of Technology*
Wuhan, China
leyangmail@163.com

3rd Zhanhui Jiang
*Chemical Machinery Equipment Manufacturing and Installation Co., Ltd,*
*Hubei Yihua Group*
Yichang, China
312679440@qq.com

4th Zhixia Ding
*School of Electrical and Information Engineering,*
*Wuhan Institute of Technology*
Wuhan, China
zxding89@163.com

5th Sai Li
*School of Electrical and Information Engineering,*
*Wuhan Institute of Technology*
Wuhan, China
sli@wit.edu.cn

*Abstract*—Nonlinear activation function is a type of function that operates within artificial neural networks, introducing nonlinearity into the network, which enables the network to be applied to a variety of nonlinear models. In the design of nonlinear activation function circuits for memristive neural networks, the main approach involves using three-piecewise linear activation functions to approximate the sigmoid and tanh functions. This paper proposes a seven-piecewise linear activation function to fit the sigmoid and tanh functions. The circuit is divided into three modules: Signal sending module converts the input voltage signal into line segments with different slopes; Signal processing module is used to resolve the issue of voltage signal discontinuities. The output voltage from Signal processing module is then processed by Signal output module for final output, which is subsequently transformed into sigmoid and tanh functions. PSPICE simulation was used to verify the correctness of the design. Subsequently, the proposed seven-piecewise linear activation function was incorporated into an iris classification network. The effectiveness of the design was demonstrated by the recognition rate of the iris classification task.

*Index Terms*—Nonlinear activation function, memristive neural networks, artificial neural networks, iris classification

## I. INTRODUCTION

Artificial neural networks are a type of mathematical model based on algorithms that mimic the characteristics of biological neural networks to carry out distributed and parallel information processing [1]. In recent years, with the rapid advancement of technology, artificial neural networks have been applied in numerous fields of life [2]–[4]. Currently, artificial neural networks primarily operate on devices with the von Neumann architecture [5], where memory and computation are separate. This leads to the need for frequent data access during processing, which significantly reduces efficiency and also results in higher power consumption. Against this backdrop, finding ways to break through the bottleneck of the von Neumann architecture and achieve low-power, highly parallel computing similar to brain-like intelligence has become increasingly important [6], [7]. In 1971, Professor Leon O. Chua from the University of California, Berkeley, proposed the concept of a memristor based on the theory of circuit symmetry [8]. It is considered to be the fourth fundamental circuit element, in addition to the resistor, capacitor, and inductor. In 2008, Strukov at HP Labs first fabricated a physical memristor based on TiO2 [9]. Due to the nanoscale size, non-volatility, and integrated memory and computation features of memristors [10]–[12], they are considered an effective method for addressing the von Neumann bottleneck of separated memory and computation. Since then, the application of memristors in artificial neural networks has become increasingly widespread [13].

In artificial neural networks, if nonlinear activation functions are not used, the network's output will only perform a linear transformation on the input, which greatly limits the network's ability to process matters [14]. Nonlinear activation functions can introduce nonlinear factors into the network, playing a crucial role in the learning and processing of complex nonlinear matters within artificial neural networks. Therefore, it is essential to incorporate nonlinear activation functions into artificial neural networks. Commonly used nonlinear activation functions include the sigmoid function and the tanh function. Since their expressions involve exponential operations, designing circuits to implement their functional graphs is relatively difficult [15].

Binary activation functions are a common method for implementing nonlinear activation functions in circuits [16]. The

This work was supported by the National Natural Science Foundation of China under Grants 62106181 and 62176189, Graduate Innovative Fund of Wuhan Institute of Technology CX2023558.

principle is that when the input is greater than the threshold, the neuron turns on; otherwise, it turns off. When using binary activation functions, it is necessary to simplify the algorithm and the network, which weakens the expressive ability of the network. Additionally, some scholars have also utilized computer software to implement sigmoid and tanh functions [17], [18], but this method mostly relies on computers and microcontrollers for realization. Moreover, this scheme requires high-energy-consuming A/D and D/A converters, which is not conducive to on-chip implementation. In order to better implement the sigmoid and tanh functions, literature [19], [20] proposes a three-piecewise linear activation function to approximate the sigmoid and tanh functions. However, this method cannot accurately fit the sigmoid and tanh functions. To the best of the author's knowledge, no scholar has yet proposed a piecewise linear activation function circuit design with more than three segments. This study introduces a seven-piecewise linear activation function circuit, which, compared to the three-piecewise linear activation function, can more accurately approximate the sigmoid and tanh functions.

The remainder of the paper is structured as follows: Section II introduces the fitting method. Section III presents the circuit and simulation of the seven-piecewise linear activation function fitting the sigmoid function. Section IV introduces the circuit and simulation of the seven-piecewise linear activation function fitting the tanh function. Section V constructs an iris classification network. Section VI summarizes the entire paper.

## II. EXPLANATION OF THE PRINCIPLE

In artificial neural networks, the widely used nonlinear activation functions include the sigmoid function and the tanh function. Taking the sigmoid function as an example, due to the continuously changing slope of the sigmoid function, it is difficult to design circuits to directly implement the sigmoid function curve. To this end, this paper designs a seven-piecewise linear activation function to approximate the sigmoid function. Compared to the three-piecewise linear activation function, the seven-piecewise linear activation function can better capture the characteristics of the sigmoid function. Fig 1(a) illustrates a schematic of the seven-piecewise linear activation function approximating the sigmoid function, where the blue line segments represent the fitting line segments, and the red line segments represent the sigmoid function itself. The fitting line segments are divided into seven parts: the first segment is a line segment with an output of 0, the second segment has a slope of $a1$, the third segment has a slope of $a2$, the fourth segment has a slope of $a3$, the fifth segment has the same slope as the third segment, the sixth segment has the same slope as the second segment, and the seventh segment is a line segment with an output of 1. Fig 1(b) is a schematic diagram of a seven-piecewise linear activation function fitting the tanh function.

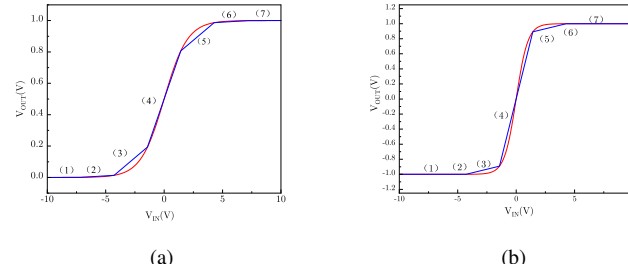

(a)  (b)

Fig. 1: Fitting diagram:(a) Schematic diagram of fitting sigmoid function with seven-piecewise linear activation function, (b) Schematic diagram of fitting tanh function with seven-piecewise linear activation function.

## III. DESIGN OF A SEVEN-PIECEWISE LINEAR ACTIVATION FUNCTION CIRCUIT FOR FITTING SIGMOID FUNCTION

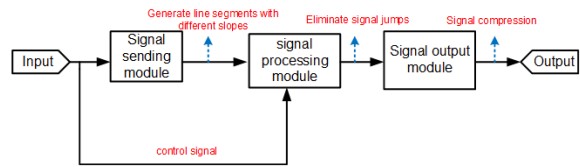

Fig. 2: Overall Structure Diagram.

The circuit structure diagram of the seven-piecewise linear activation function for fitting the sigmoid function is shown in Fig 2. The entire circuit is divided into three modules: signal sending module, signal processing module, and signal output module, with the three modules connected in sequence. The input of the circuit is a voltage signal that increases linearly with time. After passing through signal sending module, the signal can generate line segments with different slopes, But at the slope change, the signal will experience a jump. Subsequently, the output voltage from Signal sending module is processed by Signal processing module to eliminate the jump, and finally, the signal is output through Signal output module.

### A. Signal sending module

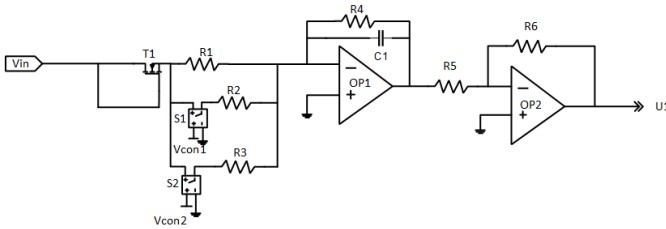

Fig. 3: Signal sending module circuit.

Signal sending module is shown in Fig 3. $V_{IN}$ is a voltage signal that changes linearly over time, ranging from 0V to

7V within 0-7$s$. T1 is an nmos transistor with an internal threshold voltage of 1V. S1 and S2 are two voltage-controlled switches with an open voltage of 2V each. OP1 and OP2 are two operational amplifiers, $R_1$=10$K\Omega$, $R_2$= $R_4$=20$K\Omega$, $R_3$=2.85$K\Omega$, $R_5$=$R_6$=1$K\Omega$,C1=20$nf$, its function is to eliminate the self-oscillation of OP1. $V_{CON1}$ and $V_{CON2}$are two control voltages. The circuit workflow is as follows:

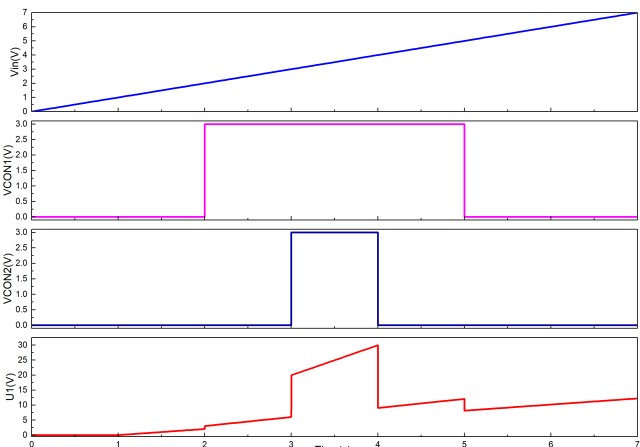

Fig. 4: Simulation of signal sending module circuit.

1)When $V_{IN}$ is less than the threshold voltage of T1, T1 turns off, and U1 outputs 0, which corresponds to line segment 1 in Fig 1(a).

2) When $V_{IN}$ is greater than the threshold of T1, the subsequent circuit has voltage input. If both $V_{CON1}$ and $V_{CON2}$ are at a low level, $V_{IN}$ becomes a negative voltage after being amplified by OP1 and then becomes a positive voltage after passing through OP2. At this time, the slope of the output voltage U1 is the amplification factor of OP1, which is the ratio of $R_4$ to $R_1$. This corresponds to line segment 2 with slope $a1$ in Fig 1 (a).

3) As $V_{IN}$ continues to increase, $V_{CON1}$ becomes high level, and the amplification factor of OP1 changes to the ratio of $R_4$ in parallel with $R_1$ and $R_2$. The slope of the output voltage U1 increases, which corresponds to line segment 3 with slope $a2$ in Fig 1 (a). When the slope of U1 changes from $a1$ to $a2$, the output voltage U1 will jump upwards.

4) As $V_{IN}$ continues to increase, $V_{CON2}$ becomes high level, and the amplification factor of OP1 changes to the ratio of $R_4$ in parallel with $R_1$, $R_2$, and $R_3$. The slope of the output voltage U1 changes again, with this slope corresponding to line segment 4 with slope $a3$ in Fig 1 (a). Similarly, the output voltage U1 will jump upwards.

5) As $V_{IN}$ continues to increase. When $V_{CON2}$ becomes low level, the amplification factor of OP1 is reset to the ratio of $R_4$ in parallel with $R_1$ and $R_2$. The slope of U1 corresponds to line segment 5 with slope $a2$ in Fig 1(a). Since $a2$ is less than $a3$, the output voltage U1 will jump downwards.

6) As $V_{IN}$ continues to increase. When $V_{CON1}$ also becomes low level, the amplification factor of OP1 finally changes to the ratio of $R_4$ to $R_1$. At this point, the slope

of the output voltage U1 corresponds to line segment 6 with slope $a1$ in Fig 1(a). Similarly, the output voltage U1 will jump downwards.

7) $V_{IN}$ continues to increase, U1 will continue to increase.

The PSPICE simulation of the signal emission module is shown in Fig 4. U1 is the output voltage of the signal emission module, and its simulation results are consistent with the theoretical analysis.

### B. Signal processing module

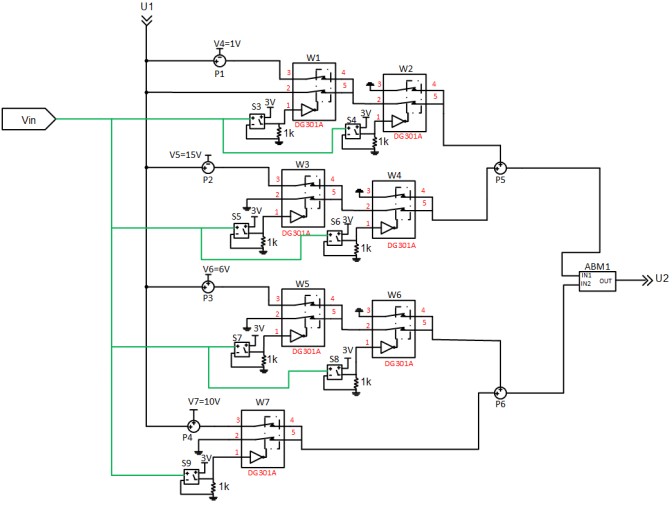

Fig. 5: Signal processing module circuit.

The role of the signal processing module is to eliminate the jump in the output signal of the signal emitting module, and its circuit schematic diagram is shown in Fig 5. U1 is the output voltage of the signal generation module, $V_{IN}$ is the input voltage, and W1-W7 are seven signal selectors of the same model. Taking W1 as an example, it has five external ports. The internal threshold voltage of the signal selector is 2.4V, Port 1 is the control port. When the voltage at $V_{port1}$ is greater than 2.4V, ports 3 and 4 of W1 are connected; otherwise, ports 2 and 5 are connected. S3-S9 are voltage-controlled switches, The switching-on voltage for S3 is 2V, for S4 and S5 it is 3V, for S6 and S7 it is 4V, and for S8 and S9 it is 5V. P1 and P2 are two subtractors, while P3-P6 are four adders. ABM1 is a digital device whose output is the sum of the voltages of IN1 and IN2. $V_4$ = 1V, $V_5$ = 15V, $V_6$ = 6V, and $V_7$ = 10V. U1 serves as the input voltage for the entire module, and $V_{IN}$ is the control voltage that governs the switching on and off of S3-S9. Taking W1 as an example, when $V_{IN}$ outputs high voltage level, S3 is turned on, causing the $V_{port1}$ of W1 to change from 0V to 3V, which results in the connection between ports 3 and 4 of W1. With the continuous input of $V_{IN}$, S3-S9 will be turned on one after another, and the workflow of the signal processing module is as follows:

1) When $V_{IN}$ is less than the T1 threshold voltage, U2 outputs 0, corresponding to the first segment of the seven-piecewise linear activation function during this stage.

2) When S3-S9 are all turned off, U1 is input from port 2 of W1 and output from port 5, and then input from port 2 of W2 and output from port 5, finally, it goes through P5 and ABM1 outputs, corresponding to the second segment of the seven-piecewise linear activation function.

3) When S3 is turned on, U1 passes through the subtractor P1, subtracts the fixed voltage $V_4$, and then input from port 3 of W1 and output from port 4, finally outputting from ABM1. This corresponds to the third segment of the seven-piecewise linear activation function.

4) When S3, S4, and S5 are turned on, ports 3 and 4 of W2 are connected, resulting in an output of 0 from W2. U1 passes through the subtractor P2, subtracts the fixed voltage $V_15$, and then input from port 3 of W3 and output from port 4 to P5, finally outputting to ABM1. This corresponds to the fourth segment.

5) When S3, S4, S5, S6, and S7 are all open, the output voltages of W2 and W4 are both 0V. U1 passes through the adder P3, adds the fixed voltage $V_6$, and then input from port 3 of W5 and output from port 4 to P6, finally outputting to ABM1. This corresponds to the fifth segment.

6) With S3-S9 all open, the outputs of W2, W4, and W6 are all 0. U1 passes through P4, adds the fixed voltage $V_7$, and then input from port 3 of W7 and output from port 4 to P6, finally outputting to ABM1. This corresponds to the sixth segment of the seven-piecewise linear activation function.

7) The operating voltage of W1-W7 is set at [-20V,20V]. As $V_{IN}$ continues to increase, U2 will remain at 20V.

*C. Signal output module*

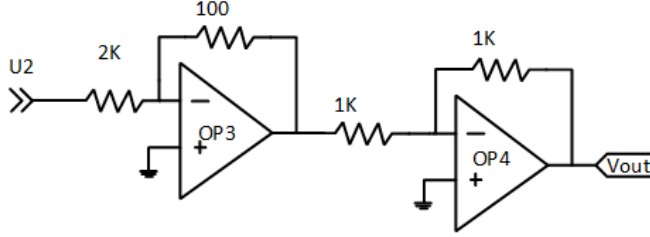

Fig. 6: Signal output module circuit.

The mapping range of the sigmoid function is [0,1]. Therefore, it is necessary to compress the output of the signal processing module. The circuit of the signal output module is shown in Fig 6. U2 is compressed by 20 times after passing through OP3, and then it is converted into a positive voltage through OP4. $V_{out}$ is the output of the entire circuit.

*D. Complete circuit*

The circuit of the seven-piecewise linear activation function fitting the sigmoid function is shown in Fig 7. The signal generation module, signal processing module, and signal output module are interconnected, and have been verified using PSPICE.

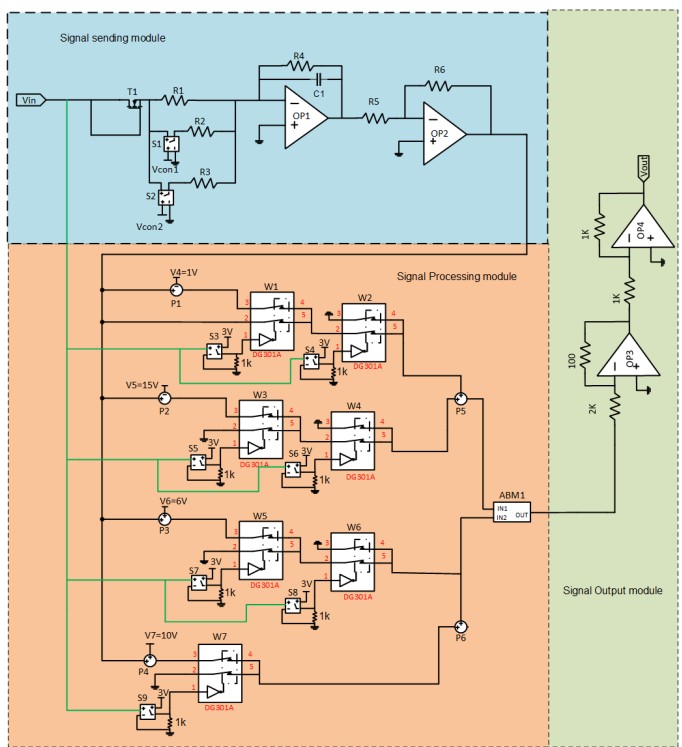

Fig. 7: Complete circuit.

*E. PSPICE simulation analysis of seven-piecewise linear activation function fitting sigmoid function*

The simulation of the seven-piecewise linear activation function circuit fitting the sigmoid function is shown in Fig 8. The simulation time is set to 7$s$. $V_{IN}$ is the input voltage of the entire circuit, and $V_{CON1}$, $V_{CON2}$ are the control voltages, as shown in Fig 4.

Between 0-1$s$, $V_{IN}$ is always less than 1V, T1 is disconnected, the subsequent circuit has no input, and ultimately $V_{out}$ is 0.

Between 1-2$s$, $V_{IN}$ is greater than 1V, $V_{IN}$ is connected to the subsequent circuit, $V_{T1}$ is the source voltage of MOSFET T1, and the output voltage after amplification and conversion by OP1 and OP2 is:

$$V_{OP2} = -\frac{R_6}{R_5}(-\frac{R_4}{R_1}V_{T1}) = 2V_{T1} \tag{1}$$

In the signal processing module, S1-S9 are all turned off, so the IN1 segment input of ABM1 is $2V_{T1}$, and the IN2 segment input is 0. The output of ABM1 is:

$$V_{ABM1} = V_{IN1} + V_{IN2} = 2V_{T1} \tag{2}$$

The signal is then sent to the signal output module for output. The relationship between the circuit's output and $V_{T1}$ is as follows:

$$V_{out} = \frac{1}{20}V_{ABM1} = \frac{1}{10}V_{T1} \tag{3}$$

Between 2-3$s$, during this phase, $V_{CON1}=3V$, S1 is turned on, and the output of OP2 is:

$$V_{OP2} = \frac{R_4(R_1+R_2)}{R_1R_2}V_{T1} = 3V_{T1} \qquad (4)$$

In the signal processing module, with S3 opened, the ports 3 and 4 of W1 are connected, and the output voltage of ABM1 is:

$$V_{ABM1} = V_{OP2} - V_4 = 3V_{T1} - 1 \qquad (5)$$

The relationship between the final circuit output and $V_{T1}$ is as follows:

$$V_{out} = \frac{1}{20}V_{ABM1} = \frac{3V_{T1}-1}{20} \qquad (6)$$

Between 3-4$s$, both $V_{CON1}$ and $V_{CON2}$ are at 3V, S1 and S2 are both activated, and the output voltage of OP2 is:

$$V_{OP2} = \frac{R_4(R_2R_3+R_1R_3+R_1R_2)}{R_1R_1R_3} = 10V_{T1} \qquad (7)$$

In the signal processing module, with S3, S4, and S5 conductive, the output voltage of ABM1 is:

$$V_{ABM1} = V_{OP2} - V_5 = 10V_{T1} - 15 \qquad (8)$$

The relationship between the final circuit output and $V_{T1}$ is:

$$V_{out} = \frac{1}{20}V_{ABM1} = \frac{2V_{T1}-3}{4} \qquad (9)$$

Between 4-5$s$, $V_{CON2}$ returns to 0V, the switch S2 is turned off, and the output voltage of OP2 is:

$$V_{OP2} = \frac{R_4(R_1+R_2)}{R_1R_2}V_{T1} = 3V_{T1} \qquad (10)$$

In the signal processing module, with S3, S4, S5, S6, and S7 all open, the output voltage of ABM1 is:

$$V_{ABM1} = V_{OP2} + V_6 = 3V_{T1} + 6 \qquad (11)$$

The final output of the circuit is:

$$V_{out} = \frac{1}{20}V_{ABM1} = \frac{3V_{T1}+6}{20} \qquad (12)$$

Between 5-6$s$, $V_{CON1}$ becomes 0V, S1 is closed, and the output voltage of OP2 is:

$$V_{OP2} = -\frac{R_6}{R_5}(-\frac{R_4}{R_1}V_{T1}) = 2V_{T1} \qquad (13)$$

In the signal processing module, with S3-S9 all open, the output voltage of ABM1 is:

$$V_{ABM1} = V_{OP2} + V_7 = 2V_{T1} + 10 \qquad (14)$$

The final output of the circuit is:

$$V_{out} = \frac{1}{20}V_{ABM1} = \frac{V_{T1}+5}{10} \qquad (15)$$

Between 6-7$s$, due to the compression of the operational amplifier, the final output of the circuit will remain at 1V.

The circuit simulation of seven-piecewise linear activation function fitting the sigmoid function is shown in Fig 8. The simulation results are consistent with the theoretical analysis, which proves the correctness of the circuit.

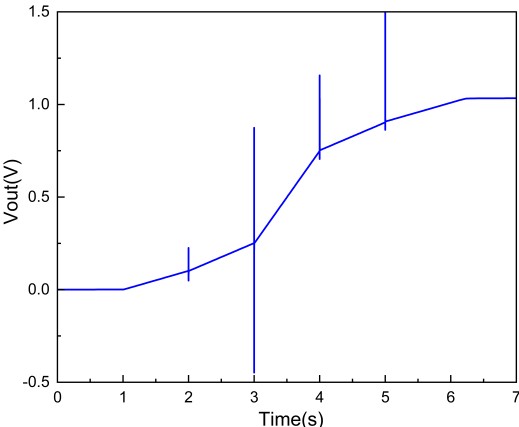

Fig. 8: Simulation of seven-piecewise linear activation function circuit fitting sigmoid function.

## IV. DESIGN OF A SEVEN-PIECEWISE LINEAR ACTIVATION FUNCTION CIRCUIT FOR FITTING TANH FUNCTION

The tanh function is quite similar to the sigmoid function, but while the sigmoid function compresses the input signal to a range between 0 and 1, the tanh function compresses the input signal to a range between -1 and 1. For this reason, we have improved the output module in Fig 6 to achieve an output range between -1 and 1. The improved signal output module is shown in Fig 9. Unlike Fig 6, in the improved circuit, a bias circuit composed of A1 is added after OP4, and the final output of the circuit is:

$$V_{\text{out}} = \frac{1}{10}U_2 - 1 \qquad (16)$$

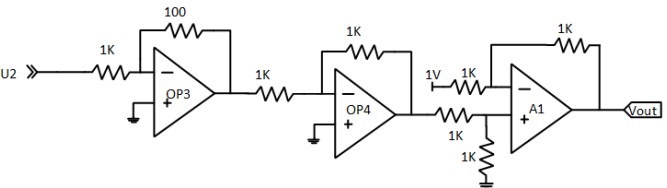

Fig. 9: The circuit for the signal output module in a seven-piecewise linear activation function circuit that fits the tanh function.

The signal sending module and the signal processing module that fit the tanh function are the same as those in Fig 3 and Fig 5. By substituting equations 2, 5, 8, 11, and 15 into equation 16, we can obtain the middle five segments of the seven-piecewise linear function that fits the tanh function. The analysis of the first segment and the last segment is the same as that of the seven-piecewise linear activation function fitting the sigmoid function. The PSPICE simulation of the seven-piecewise linear function fitting the tanh function is shown

in Fig 10. The simulation is consistent with the theoretical analysis, which proves the effectiveness of the circuit.

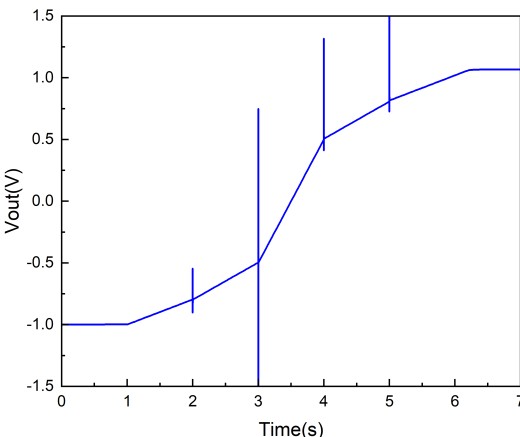

Fig. 10: Simulation of seven-piecewise linear activation function circuit fitting tanh function.

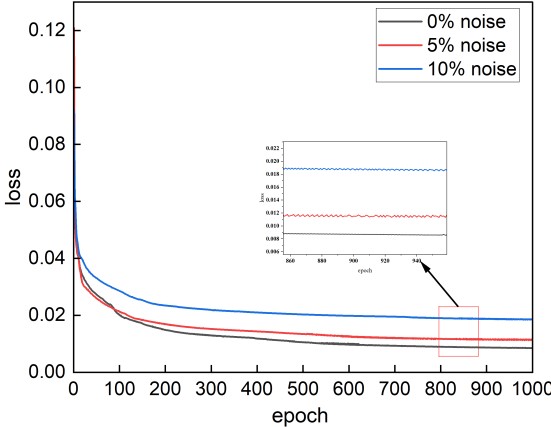

Fig. 11: Training loss.

## V. EXPERIMENT AND ANALYSIS

### A. Iris classification experiment

To verify the effectiveness of this circuit in memristive neural networks, an iris classification network was constructed . The network consists of two layers, with the number of neurons in the input layer, hidden layer, and output layer being 4, 8, and 1, respectively. The initial weights of the network are between 0 and 0.25. We incorporated both the three-piecewise linear activation function and the seven-piecewise linear activation function fitting the sigmoid function into the network as activation functions. The experiment was repeated ten times, and the final results were taken as the average of the ten runs. The experimental results show that the iris recognition rate of the network using the three-piecewise linear activation function is 80.22%, while the iris recognition rate of the network using the seven-piecewise linear activation function is 94%. The experiment demonstrates that the performance of the network using the seven-piecewise linear activation function is superior to that of the network using the three-piecewise linear activation function. To test the stability of the network using the seven-piecewise linear activation function, we added 5% and 10% noise to the network. The training errors are shown in Fig 11, which proves that the network has good robustness against noise.

### B. Monte carlo and temperature analysis

In order to test the impact of circuit component errors on circuit performance, we conducted a Monte Carlo analysis on the seven-piecewise linear activation function circuit for fitting the sigmoid function. We set the tolerances of $R_4$ in Fig 7 to 10% respectively, and set the number of simulations to 100

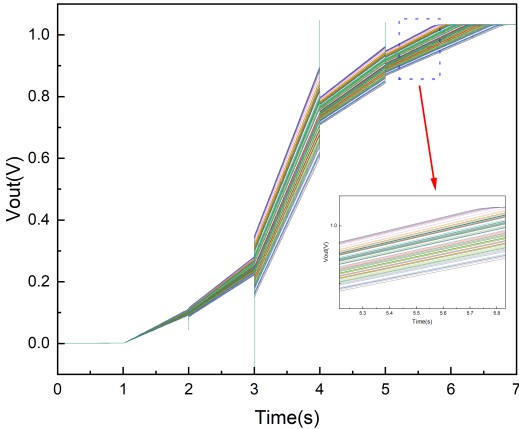

Fig. 12: Monte carlo analysis.

times. The simulation results are shown in Fig 12 , which demonstrates that the circuit has high robustness.

In practical circuits, temperature can affect the parameter values of electronic devices. To simulate the impact of temperature on the circuit proposed in this paper, a temperature analysis was conducted on the circuit shown in Fig 7. Fig 12 illustrates the simulation results of the circuit at different temperatures: 0°C, 10°C, 20°C, 30°C, 50°C, 80°C, and 100°C. The simulation analysis shows that the seven-piecewise linear activation function fitting the sigmoid function proposed in this paper can operate normally within a temperature range of 0-100°C.

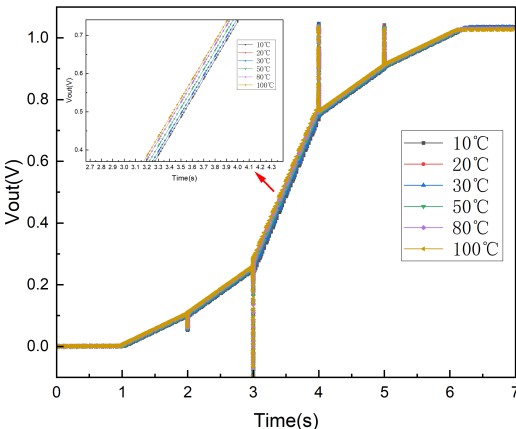

Fig. 13: Temperature analysis simulation diagram.

## VI. CONCLUSION

This paper constructs a seven-piecewise linear activation function circuit for memristive neural networks, which can fit the sigmoid function and the tanh function, respectively. Firstly, the circuit structures and working processes of the three modules in the seven-piecewise linear activation function circuit for fitting the sigmoid function are introduced, followed by simulations of the overall circuit. Subsequently, a bias circuit is added to the seven-piecewise linear activation function circuit that fits the sigmoid function, which enables the seven-piecewise linear activation fitting of the tanh function. Subsequently, to verify the effectiveness of the seven-piecewise linear activation function, we constructed an iris classification network. We incorporated both three-piecewise and seven-piecewise linear activation functions into the network separately. In the end, we found that the network using the seven-piecewise linear activation function achieved a higher recognition rate for iris flowers. Finally, we conducted Monte Carlo analysis and temperature analysis on the seven-piecewise linear activation function fitting the sigmoid function. The simulation results indicate that the circuit has good stability.

## ACKNOWLEDGMENT

This work was supported by the National Natural Science Foundation of China under Grants 62106181 and 62176189, Graduate Innovative Fund of Wuhan Institute of Technology CX2023558.

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
