# OpenReview forum: "Circuit Design of a Seven-piecewise Linear  Activation Function"
_IEEE.org/ICIST/2024/Conference — IEEE ICIST 2024 Conference Submission_

### Official Review · Reviewer_kVsX · 2024-08-25
**Accept, some mistakes should be revised**

**Rating:** 7
**Confidence:** 3

**Review:**

1. The highlights of this paper should be added before the structural description in the Introduction.
2. The font size of the illustration should be smaller than the body of the paper.
3. The demonstrations of the test system in Section V is insufficient.

---

### Official Review · Reviewer_jqsU · 2024-08-30
**The paper is interesting**

**Rating:** 7
**Confidence:** 3

**Review:**

1. What motivated the choice of a seven-piecewise linear activation function over the commonly used three-piecewise linear functions? How does this design improve the approximation of the sigmoid and tanh functions compared to previous approaches?
2. The abstract mentions the use of the proposed activation function in an iris classification network. Could the authors provide more details on how the seven-piecewise linear activation function specifically impacted the recognition rate or overall performance compared to traditional activation functions used in similar tasks?

---

### Official Review · Reviewer_St16 · 2024-08-31
**This paper proposes a seven-piecewise linear activation function to fit the sigmoid and tanh functions. The proposed strategy is attractive for an iris classification network.**

**Rating:** 6
**Confidence:** 5

**Review:**

1.Abstract section does not describe the problem and motivation clearly.
2.It is recommended to increase the comparison of results to highlight the advantages of research methods.
3.Designing nonlinear activation function circuits for memristive neural networks requires a multidisciplinary approach, integrating insights from circuit design, materials science, and neural network theory to develop effective and reliable activation function circuits for memristive neural networks. How do the authors consider this problem?
4.Grammatical errors, such as ‘In order to better implement the sigmoid and tanh functions, literature [19], [20] proposes a three-piecewise linear activation function to approximate the sigmoid and tanh functions.’ It is advisable to carefully review the entire text.
5.The ‘SEP’ in reference [1] and the ‘SEPT’ in reference [13] are inconsistent. It is advisable to carefully check.

---

### Decision · Program_Chairs · 2024-09-06

Accept (Oral)